# Structure of RNA polymerase bound to ribosomal 30S subunit

Gabriel Demo[1†], Aviram Rasouly[2,3†], Nikita Vasilyev[2], Vladimir Svetlov[2], Anna B Loveland[1], Ruben Diaz-Avalos[4], Nikolaus Grigorieff[4], Evgeny Nudler[2,3*], Andrei A Korostelev[1*]

[1]RNA Therapeutics Institute, Department of Biochemistry and Molecular Pharmacology, University of Massachusetts Medical School, Worcester, United States; [2]Department of Biochemistry and Molecular Pharmacology, New York University School of Medicine, New York, United States; [3]Howard Hughes Medical Institute, New York University School of Medicine, New York, United States; [4]Janelia Research Campus, Howard Hughes Medical Institute, Ashburn, United States

**Abstract** In bacteria, mRNA transcription and translation are coupled to coordinate optimal gene expression and maintain genome stability. Coupling is thought to involve direct interactions between RNA polymerase (RNAP) and the translational machinery. We present cryo-EM structures of *E. coli* RNAP core bound to the small ribosomal 30S subunit. The complex is stable under cell-like ionic conditions, consistent with functional interaction between RNAP and the 30S subunit. The RNA exit tunnel of RNAP aligns with the Shine-Dalgarno-binding site of the 30S subunit. Ribosomal protein S1 forms a wall of the tunnel between RNAP and the 30S subunit, consistent with its role in directing mRNAs onto the ribosome. The nucleic-acid-binding cleft of RNAP samples distinct conformations, suggesting different functional states during transcription-translation coupling. The architecture of the 30S•RNAP complex provides a structural basis for co-localization of the transcriptional and translational machineries, and inform future mechanistic studies of coupled transcription and translation.
DOI: https://doi.org/10.7554/eLife.28560.001

**\*For correspondence:**
evgeny.nudler@nyumc.org (EN);
andrei.korostelev@umassmed.edu
(AAK)

[†]These authors contributed
equally to this work

**Competing interest:** See
page 13

**Reviewing editor:** Rachel
Green, Johns Hopkins School of
Medicine, United States

## Introduction

In bacteria, coupling of transcription and translation coordinates optimal gene expression and maintains genome stability (*McGary and Nudler, 2013*). Coupling is thought to involve direct interactions between RNA polymerase (RNAP) and the translational machinery to allow cellular co-localization for efficient translation initiation (*Sanamrad et al., 2014*) and to prevent RNAP from backtracking during transcription elongation (*Proshkin et al., 2010*).

Bacterial RNAP's functional core comprises five proteins, sufficient for transcription elongation. These include two α-subunits, which form the foundation for recruitment of the large β- and β'-subunits and a small ω-subunit (*Finn et al., 2000*; *Murakami, 2015*). The β- and β'- subunits form two halves of the crab-claw-shaped core, enclosing the nucleic acid-binding cleft (*Zhang et al., 1999*; *Werner and Grohmann, 2011*; *Sekine et al., 2015*; *Kang et al., 2017*). The ribosome consists of the small (30S) and large (50S) subunits. The 30S subunit initiates protein synthesis by binding mRNA and initiator tRNA (*Simonetti et al., 2009*). Placement of the mRNA open-reading frame (ORF) is facilitated by base-pairing of the ORF-preceding Shine-Dalgarno sequence with the 3′-tail of the 16S ribosomal RNA (rRNA) of the 30S subunit (*Shine and Dalgarno, 1975*; *Steitz and Jakes, 1975*; *Hui and de Boer, 1987*; *Jacob et al., 1987*).

Early genetic studies suggested interactions between RNAP and the 30S subunit in *E. coli* (*Chakrabarti and Gorini, 1977*) and *M. smegmatis* (*Karunakaran and Davies, 2000*). More recent work showed that RNAP co-purifies with ribosomal proteins S1 (*Sukhodolets and Garges, 2003*) and S2 (*Büttner et al., 1997*), consistent with a direct interaction between RNAP and the small ribosomal subunit in the cell. S1 enhances the transcription rate of RNAP (*Sukhodolets et al., 2006*) and is implicated in loading mRNA onto the translation machinery (*Duval et al., 2013*), suggesting that S1 participates in transcription-translation coupling. To bring insights into the coupling of transcription and translation, we studied in this work how RNAP interacts with the ribosome and ribosomal subunits. We find that RNAP binds the 30S subunit with high affinity and report cryo-EM structures of the 30S•RNAP complex.

## Results

### RNAP binds the 30S subunit

To investigate whether RNAP directly interacts with the translational machinery, we tested whether the functional *E. coli* RNAP core complex binds the *E. coli* 30S or 50S ribosomal subunits or the 70S ribosome. Using sucrose gradient fractionation and dot-blot assays, we found that RNAP binds stably to the 30S subunit, less to the 50S subunit, and little RNAP binds to the 70S ribosome (*Figure 1—figure supplement 1*). The 30S•RNAP complex is stable under a wide range of conditions, including those that mimic ionic conditions in living cells and optimally couple transcription and transcription in vitro (*Figure 1—figure supplement 2*) (*Jewett and Swartz, 2004*). The binding affinity ($K_d$) of $\leq$50 nM under cell-like ionic conditions (see *Figure 1—figure supplement 3*, and Materials and methods) indicates high stability of the complex at cellular concentrations of free RNAP ($\sim$1 $\times$ 10$^{-6}$ M) (*Klumpp and Hwa, 2008*) and ribosomes/subunits (50–100 $\times$ 10$^{-6}$ M) (*Bakshi et al., 2012*). Bis-(sulfosuccinimidyl)-suberate cross-linking mapped by mass spectrometry revealed that RNAP $\beta$- and $\beta'$- subunits are in close proximity to 30S ribosomal proteins S1, S2, S18, and S21 (*Table 1*), which enclose the mRNA-binding region of the 30S subunit, near the 3' end of 16S rRNA. Our findings therefore indicate that RNAP forms specific interactions with the 30S subunit.

### Cryo-EM structures of the 30S•RNAP complex

We next visualized the 30S•RNAP complex using single-particle electron cryo-microscopy (cryo-EM) at $\sim$7 Å resolution. Strong density shows RNAP docked near the mRNA-binding site of the 30S subunit between the head and platform domains (*Figure 1*; *Figure 1—figure supplements 4* and *5*). We also observe weak density next to RNAP consistent with a second sub-stoichiometric or mobile monomer of RNAP (*Figure 1—figure supplement 5A–B*). The position of the second RNAP monomer coincides with that in RNAP dimers seen in previous studies (*Zhang et al., 1999*; *Kansara and Sukhodolets, 2011*), placing the second RNAP monomer more than 50 Å from the binding site between the primary RNAP monomer and the 30S subunit.

Consistent with our cross-linking results, the $\beta'$- and $\beta$-subunits of RNAP form two primary contacts with the 30S subunit (*Figure 2A–C*; *Figure 2—figure supplement 1*). The N-terminal Zn-finger domain of the $\beta'$ subunit (*Figure 2B*) interacts extensively with ribosomal proteins S2 and S21 and with hairpin-loop 40 (h40) of 16S rRNA. The positively-charged tip of the Zn-finger motif (aa 75–85) binds between a $\beta$-sheet and $\alpha$-helices of S2, which exposes conserved Asp188, Asp204, and Asp205. Hydrophobic side chains of Leu78 and Ile84 from the $\beta'$ subunit pack near Phe16 and Ile207 of S2, respectively. The Zn-bound surface is placed near the C-terminus of S21 and the tip of h40 at U1168. An $\alpha$-helix (aa 264–284) of the $\beta'$ subunit near the Zn-binding motif is tilted so that Asp284 contacts the ribose of A1163 of 16S rRNA at h40. The second binding site between RNAP and 30S is formed by hydrophobic packing of the $\beta$-subunit flap helix ($\beta$-flap) residues Leu901, Leu902, Ile905, and Phe906 onto S18 residues Phe10, Phe13, Val40, and Ile44 (*Figure 2C*). The area of contact between RNAP and the 30S subunit is $\sim$1750 Å$^2$, indicating that RNAP and 30S form a stable functional complex (*Jones and Thornton, 1996*; *Bogan and Thorn, 1998*; *Day et al., 2012*).

Maximum likelihood classification reveals conformational rearrangements of RNAP adopting two conformations on the 30S subunit (*Figure 1—figure supplement 5*). The two states are related to each other by $\sim$7° rotation around an axis positioned near the Zn-finger motif (*Figure 2—figure supplement 2A*). The rotation widens the nucleic-acid-binding cleft of RNAP, as the $\beta$ and $\beta'$ subunits

**Table 1.** In vitro BS3 X-links between subunits of E. coli RNA polymerase and 30S ribosome subunit.

Positions of X-linked residues are numbered according to full polypeptide sequences of corresponding proteins (column A) or sequences of X-linked peptides (1 being the N-terminal residue of the each peptide) (column C). Lower e-value indicates higher confidence of computational discovery of a given X-link (column B). In cases where several precursors for a given X-link were discovered (column F) the sequence of the X-linked peptide (column C), theoretical mass (column D), and experimental deviation from the theoretical mass (column E) are shown for the precursor with the lowest e-value (column B).

| X-link | Lowest e-value | Sequence of X-linked peptides and location of X-linked residues | Theoretical precursor mass (Da) | Deviation from theoretical mass (Da) | Number of precursors |
|---|---|---|---|---|---|
| RpoB(900) RS1(450) | 4.65E-010 | GETQLTPEEKLLR(10)-KGAIVTGK(1) | 2423.363414 | 0.99628 | 1 |
| RpoC(87) RS1 (450) | 2.85E-007 | GVICEKCGVEVTQTK(6)-KGAIVTGK(1) | 2617.381784 | 2.99751 | 1 |
| RpoC(13) RS2 (11) | 6.73E-007 | AQTKTEEFDAIK(4)-DMLKAGVHFGHQTR(4) | 3113.560294 | 3.00782 | 1 |
| RpoB(900) RS18(30) | 9.38E-005 | GETQLTPEEKLLR(10)-DIATLKNYITESGK(6) | 3202.697164 | 2.00093 | 1 |
| RpoB(900) RS21(5) | 2.75E-004 | GETQLTPEEKLLR(10)-PVIKVR(4) | 2361.363024 | −0.00541 | 2 |
| RpoC(87) RS4 (185) | 1.09E-003 | GVICEKCGVEVTQTK(6)-RKPER(2) | 2529.304204 | 2.00309 | 1 |
| RpoB(265) RS19(29) | 3.12E-003 | VYVEKGRR(5)-KPLR(1) | 1655.983404 | −0.01571 | 2 |
| RpoB(476) RS1(272) | 6.47E-003 | AVKER(3)-QLGEDPWVAIAKR(12) | 2221.221774 | −0.00039 | 1 |
| RpoC(87) RS4 (185) | 7.47E-003 | GVICEKCGVEVTQTK(6)-RKPER(2) | 2529.304204 | 2.00389 | 2 |
| RpoC(50) RS21(25) | 1.36E-002 | TFKPER(3)-SCEKAGVLAEVR(4) | 2232.157124 | 0.99497 | 1 |
| RpoB(265) RS19(29) | 1.50E-002 | VYVEKGRR(5)-KPLR(1) | 1655.983404 | −0.01581 | 1 |
| RpoB(909) RS21(25) | 2.25E-002 | AIFGEKASDVK(6)-SCEKAGVLAEVR(4) | 2619.357654 | 1.00524 | 1 |
| RpoC(50) RS1 (434) | 8.29E-002 | TFKPER(3)-ISLGVKQLAEDPFNNWVALNK(6) | 3269.744714 | 4.00858 | 1 |
| RpoC(1192) RS16(80) | 8.79E-002 | LVITPVDGSDPYEEMIPKWR(18)-VAALIKEVNKAA (10) | 3707.984664 | 2.00023 | 1 |

DOI: https://doi.org/10.7554/eLife.28560.002

move apart by at least 3 Å (**Figure 2—figure supplement 2B**). Both conformations are intermediate between the open free *E. coli* RNAP core (**Darst et al., 2002**) and the closed holoenzyme (bound with σ-factor) (**Finn et al., 2000**; **Murakami et al., 2002**; **Vassylyev et al., 2002**; **Murakami, 2013**) or closed elongating RNAP (i.e., bound with DNA and RNA) (**Vassylyev et al., 2007b**, **2007a**; **Liu et al., 2015**; **Kang et al., 2017**).

Opening of RNAP is implicated in transcriptional elongation and pausing (**Hein et al., 2014**). To determine if the RNAP conformation in our structure is compatible with elongating or paused RNAP, we superimposed the crystal structure of *T. thermophilus* RNAP elongation complex (**Vassylyev et al., 2007a**) onto our 30S•RNAP structure (**Figure 3** and **Figure 3—figure supplement 1A**). The alignment shows no pronounced clashes between the domains of RNAP and 30S subunit. The flexible β-flap helix overlaps with the tip of h26, suggesting that a local rearrangement may be required. Thus, elongating RNAP can in principle interact with the 30S subunit.

## S1 is aligned with RNA paths of RNAP and 30S

Elongated density connects the β′ subunit of RNAP to the Shine-Dalgarno binding site at the 3′ tail of 16S rRNA (**Figure 2D–E**, and **Figure 1—figure supplement 5A–E**). The Shine-Dalgarno binding site lies between the head and platform of the 30S subunit, where protein S1 is thought to bind and

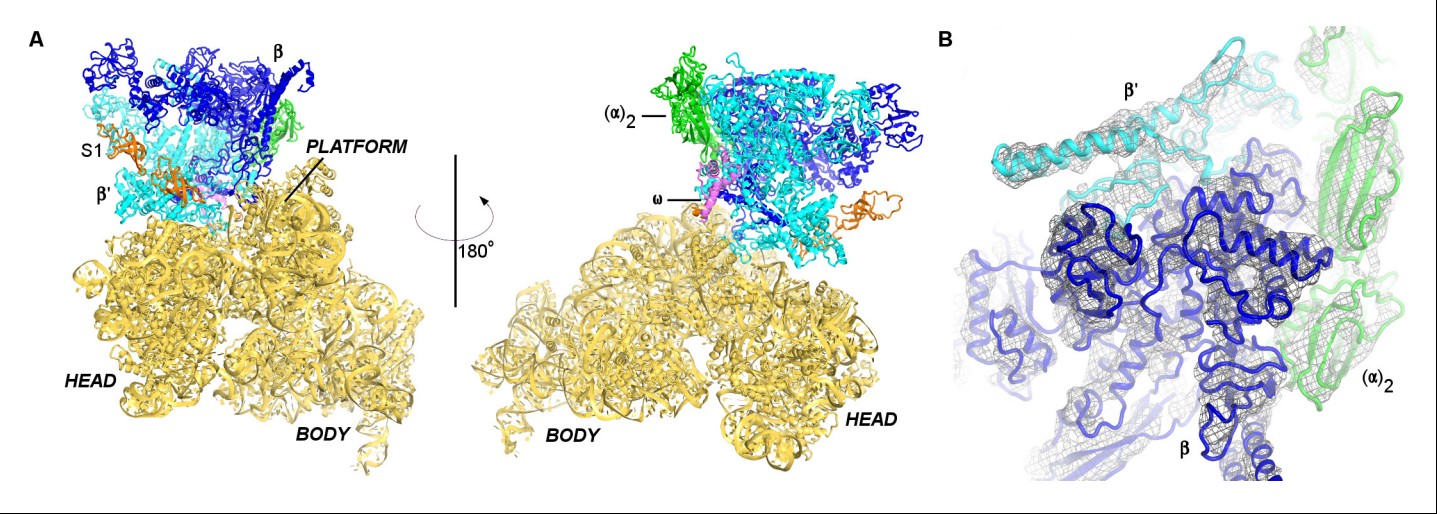

**Figure 1.** Cryo-EM structure of the 30S•RNAP complex. (**A**). The 30S•RNAP structure. The 30S subunit is in yellow (ribosomal protein S1 is in orange) with domains labeled. RNAP domains are colored in green (α-subunits), blue (β-subunit), cyan (β′-subunit) and violet (ω-subunit). (**B**) Cryo-EM density (gray mesh) for RNAP (shown at σ = 5). The view shows the top of RNAP,~90° relative to that shown in the left panel (**A**).

DOI: https://doi.org/10.7554/eLife.28560.003

The following figure supplements are available for figure 1:

**Figure supplement 1.** Binding of RNAP core to the ribosome or ribosomal subunits.

DOI: https://doi.org/10.7554/eLife.28560.004

**Figure supplement 2.** Binding of 1 μM RNAP core to the 30S ribosomal subunit in a range of cell-like conditions including glutamate and acetate anions and potassium and magnesium cations, optimized for transcription-translation coupling (*Jewett and Swartz, 2004*).

DOI: https://doi.org/10.7554/eLife.28560.005

**Figure supplement 3.** Binding of 50 nM RNAP core to the 30S subunit under cell-like conditions optimized for transcription-translation coupling (*Jewett and Swartz, 2004*).

DOI: https://doi.org/10.7554/eLife.28560.006

**Figure supplement 4.** Schematic of cryo-EM refinement and classification of the 30S•RNAP dataset.

DOI: https://doi.org/10.7554/eLife.28560.007

**Figure supplement 5.** Cryo-EM densities for the 30S•RNAP and 30SΔS1•RNAP complexes.

DOI: https://doi.org/10.7554/eLife.28560.008

facilitate recruitment of mRNA during translation initiation (*Sengupta et al., 2001*; *Lauber et al., 2012*; *Duval et al., 2013*; *Park et al., 2014*; *Byrgazov et al., 2015*). We were able to model oligo-nucleotide-binding (OB) domains 2 and 3 of S1 into this density (*Figure 2D*, and *Figure 1—figure supplement 5D*), consistent with studies showing that S1 domain 2 interacts with the 30S subunit (*Giorginis and Subramanian, 1980*; *Lauber et al., 2012*) and with our crosslinking experiments showing that aa 272 of S1 domain 3 is in close proximity to aa 420 of RNAP β-subunit (*Figure 2—figure supplement 1* and *Table 1*). Because the six OB domains of S1 are structurally similar and the resolution of our maps does not allow unambiguous domain interpretation, other OB domains of S1 could occupy this density. The globular region of OB domain 1 is not resolved in our density, but the N-terminal helix of domain 1 packs on α-helices of S2 (*Figure 1—figure supplement 5F*), as proposed previously (*Lauber et al., 2012*; *Byrgazov et al., 2015*).

To confirm that the elongated density belongs to S1, we depleted S1 from our 30S preparation (30SΔS1; *Figure 1—figure supplement 1E*), and then obtained a cryo-EM map of 30SΔS1•RNAP complex (*Figure 2E*; *Figure 1—figure supplement 5C*). The 30SΔS1•RNAP cryo-EM map shows that RNAP binds the 30S subunit similarly to that in the 30S•RNAP complex, but density connecting the β′ subunit of RNAP to the 3′ tail of 16S rRNA and density ascribed to the N-terminal helix of S1 are missing (*Figure 1—figure supplement 5D–G*). Thus, our 30S•RNAP structure visualizes S1 stabilized by RNAP.

Remarkably, the RNA exit tunnel of RNAP lies next to the Shine-Dalgarno binding site on the 30S subunit (*Figure 3A*). The 30S subunit is stabilized in an open conformation, with proteins S7 (head)

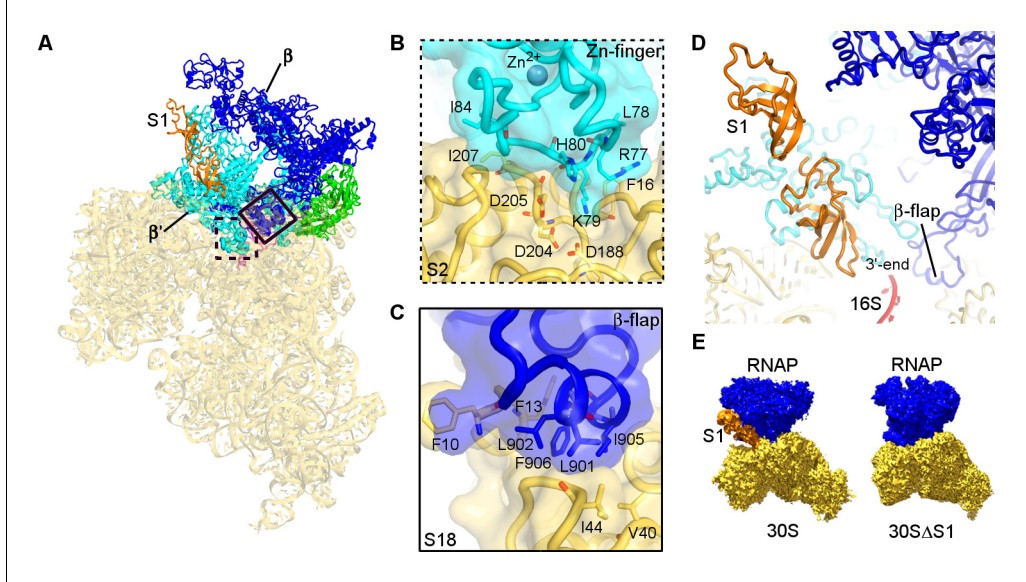

**Figure 2.** Structural basis for binding of RNAP to the 30S subunit. (**A**). Two binding sites of RNAP (boxed). (**B**) Close-up view of the Zn-finger interactions with S2. (**C**) Close-up view of the β−flap helix packing on S18. Molecular surface, secondary structure and sticks are shown in panels (**B**) and (**C**). (**D**) Position of two OB domains of S1 (orange) near the 3′ end of 16S rRNA (red). (**E**) Comparison of segmented maps of the 30S•RNAP complex formed with and without S1.

DOI: https://doi.org/10.7554/eLife.28560.009

The following figure supplements are available for figure 2:

**Figure supplement 1.** Crosslinked sites mapped on the 30S•RNAP structure.

DOI: https://doi.org/10.7554/eLife.28560.010

**Figure supplement 2.** Conformational rearrangements of RNAP between the non-rotated (blue) and rotated (gray) states.

DOI: https://doi.org/10.7554/eLife.28560.011

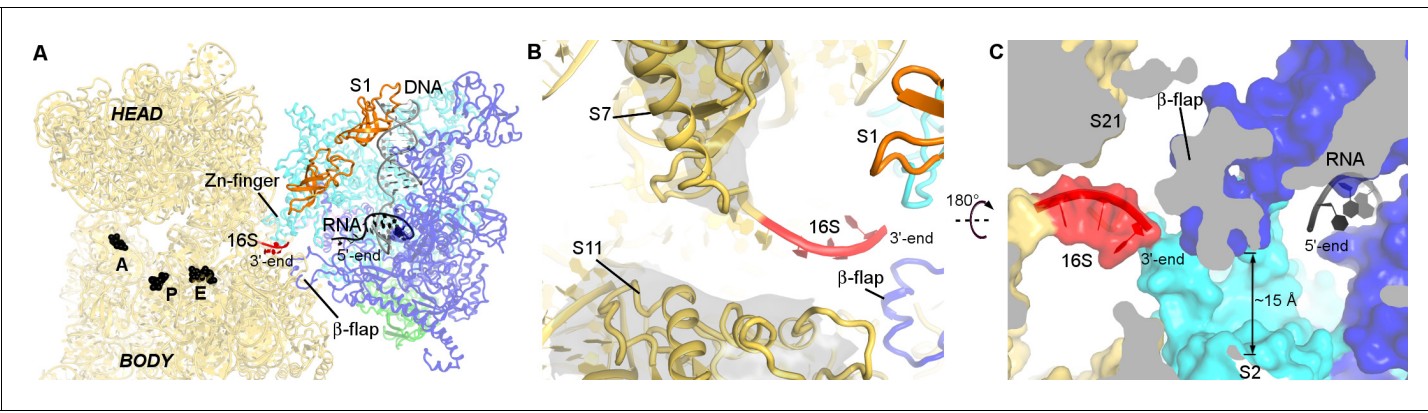

**Figure 3.** RNA exit region of RNAP is next to the SD-binding site of the 30S subunit. (**A**) Positions of DNA and RNA (gray and black), inferred from the elongation complex (*Vassylyev et al., 2007a*) (PDB 2O5I), relative to the 30S subunit. G530, A790 and G693 of 16S rRNA (black) denote the A, P and E sites on the 30S subunit. (**B**) The opening between S7 (head) and S11 (platform) of the 30S subunit. Cryo-EM density around S7 and S11 is shown (gray surface, shown at σ = 5). (**C**) RNA exit of RNAP is near the 3′-end of 16S rRNA. RNA bound to RNAP elongation complex is shown in black.

DOI: https://doi.org/10.7554/eLife.28560.012

The following figure supplement is available for figure 3:

**Figure supplement 1.** Comparison of RNAP structure in the 30S•RNAP complex with RNAP elongation complex and holoenzyme.

DOI: https://doi.org/10.7554/eLife.28560.013

and S11 (platform) separated by ~15 Å, allowing access to the mRNA path from the E to A site (*Figure 3B*). The RNA exit of RNAP at the β-flap helix connects with the 3'-end of 16S rRNA via a ~15 Å-wide opening surrounded by proteins S1, S2, S18 and S21 (*Figure 3C*). Thus, the opening between RNAP and the Shine-Dalgrano binding region of 16S rRNA could accommodate single-stranded RNA. Moreover, the location of S1 suggests that S1 could function as a fence that directs the mRNA to the Shine-Dalgarno binding site as it exits RNAP.

## Discussion

Our work uncovers a specific RNAP binding site on the 30S subunit, consistent with previously observed interactions of RNAP with S1 (*Sukhodolets and Garges, 2003*) and S2 (*Büttner et al., 1997*). Additionally, while our manuscript was under review, Fan *et al.* (*Fan et al., 2017*) reported cross-linking studies showing that RNAP binds the 30S subunit and the 70S ribosome next to the 3'-tail of the 16S ribosomal RNA, which agrees with our findings. The RNAP and 30S binding interface comprises residues that are conserved in bacteria, suggesting a conserved function involving the observed interactions.

In the cell, interactions between RNAP and the 30S subunit may occur during 30S subunit maturation or mRNA transcription-translation coupling (*Figure 4*), as well as during cellular stress or other conditions resulting in accumulation of RNAP core and ribosomal subunits. In the first scenario, our 30S•RNAP structure might represent a complex formed during transcription of rRNA operons to facilitate co-transcriptional maturation of the 30S subunit (*Figure 4A*). Direct interaction between RNAP and 30S could be important for proper 16S rRNA folding and recruitment of ribosomal proteins, including the late binders S2 and S21 (*Mizushima and Nomura, 1970*; *Culver, 2003*) and loosely associated S1. At late stages of maturation, RNAP binding to 30S could modulate rRNA transcription and facilitate pre-rRNA cleavage.

Our work suggests at least two possible mechanisms for coupling mRNA transcription and translation (*Figure 4B–E*). First, cellular co-localization of RNAP and 30S by direct binding (*Figure 4B*) could increase the probability of co-transcriptional translation initiation, and is consistent with penetration of ribosomal subunits into the nucleoid (*Bakshi et al., 2014*; *Sanamrad et al., 2014*). Because the 30S-binding site on RNAP in our structure overlaps with those of sigma factors (*Figure 3—figure supplement 1B*) (*Finn et al., 2000*; *Murakami et al., 2002*; *Vassylyev et al., 2002*;

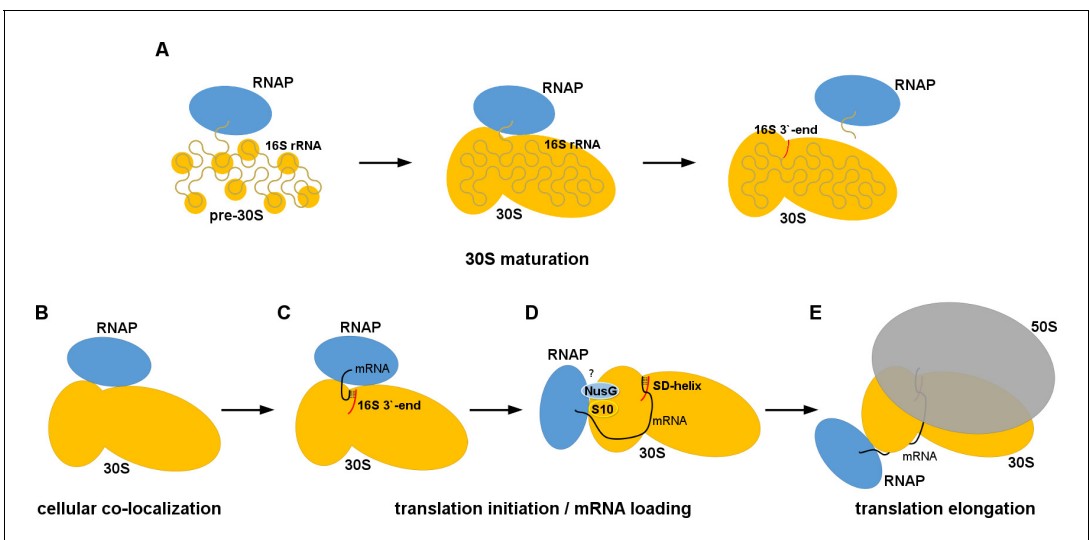

**Figure 4.** Models for 30S•RNAP complex formation in transcription-translation coupling. (**A**) Promotion of co-transcriptional maturation of the 30S subunit. Direct interaction might facilitate 16S rRNA folding, 30S protein assembly or rRNA cleavage. (**B-E**) Coupling of mRNA transcription with translation. 30S-RNAP binding might co-localize 30S and RNAP or structurally pre-arrange 30S to facilitate mRNA loading onto the 30S subunit for co-transcriptional translation initiation.

DOI: https://doi.org/10.7554/eLife.28560.014

*Murakami, 2013*), we would expect RNAP to detach from the 30S subunit during transcription initiation. Second, alignment of the RNA exit tunnel of RNAP with the Shine-Dalgarno binding site of 30S could couple transcription elongation with translation initiation (*Sanamrad et al., 2014*), by facilitating mRNA loading onto the 30S subunit (*Figure 4C*).

The affinity of RNAP to the 70S ribosome is lower than to the 30S subunit (*Figure 1—figure supplement 1*) (*Fan et al., 2017*), suggesting that mRNA or other factors are needed to strengthen binding of RNAP to translating 70S ribosome. A recent 9 Å cryo-EM structure of the mRNA-tethered 70S•RNAP complex, stabilized using glutaraldehyde crosslinking (*Kohler et al., 2017*), shows RNAP near the mRNA tunnel entry (formed by proteins S3, S4 and S5) of the trailing ribosome. This region is more than 80 Å from the binding site observed in our work (*Figure 5*). If our structure represents a co-localization or translation initiation step, then a large-scale relocation of RNAP would be required to form the 70S•RNAP elongation complex.

The likely mechanism for RNAP relocation involves dissociation of RNAP from the 30S subunit upon formation of either the Shine-Dalgarno helix or the 30S initiation complex with fMet-tRNA$^{fMet}$. Formation of the initiation complex induces a closed 30S conformation (i.e., S7 moves toward S11) (*Julián et al., 2011*) (*Figure 3B*). This rearrangement of 30S would re-structure the 30S•RNAP binding site causing RNAP to dissociate (*Figure 4D*) and ultimately dock at the mRNA tunnel entry upon mRNA accommodation in the 70S ribosome (*Figure 4E*).

Neither our structure nor the 70S•RNAP structure accounts for the role of the transcription factor NusG in transcription-translation coupling (*Zellars and Squires, 1999*). NusG bridges RNAP β'-subunit and ribosomal protein S10 (*Burmann et al., 2010*). In the mRNA-tethered 70S•RNAP complex (*Kohler et al., 2017*), the NusG-binding sites at the β'-subunit (*Liu and Steitz, 2017*) and S10 are located on the opposite ends of RNAP (*Figure 5B*). RNAP is located on opposite sides of the 30S head in our 30S•RNAP complex and in the 70S•RNAP complex, with S10 between the binding sites (*Figure 5*). Perhaps NusG interacts with S10 during the transition from translation initiation to elongation (*Figure 4D*), and helps direct RNAP toward the mRNA tunnel entry.

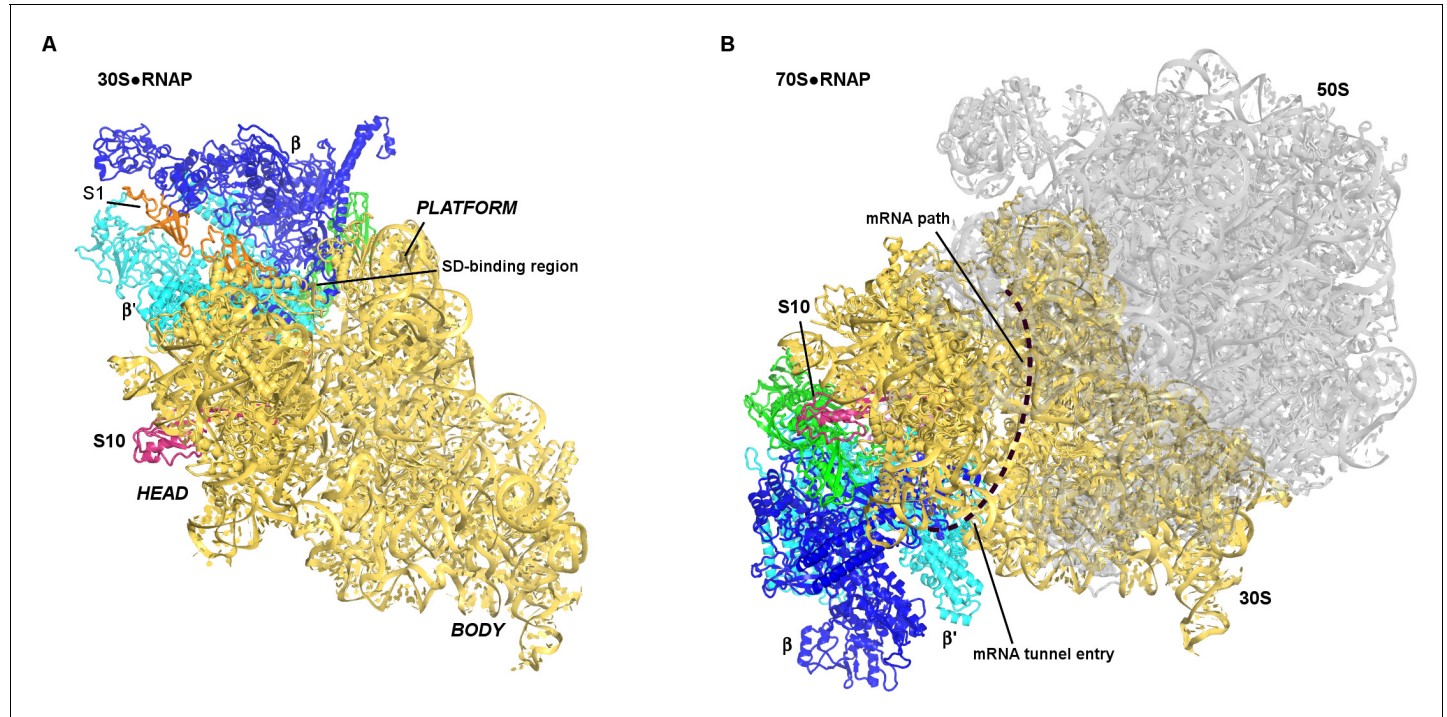

**Figure 5.** Positions of RNAP on the 30S subunit (this work, panel A) and the 70S elongation complex ([*Kohler et al., 2017*]; panel B). The views show similar orientations of the 30S subunit. The 50S subunit for the mRNA-tethered 70S•RNAP structure (PDB 5MYI) (*Kohler et al., 2017*) is shown in gray and was adapted from a high-resolution 70S structure (PDB 5U9F) (*Demo et al., 2017*). RNAP domains are colored in green (α-subunits), blue (β-subunit) and cyan (β'-subunit). Protein S10 is shown in pink.
DOI: https://doi.org/10.7554/eLife.28560.015

This study reveals specific interactions of RNAP and 30S subunit, and adds to our understanding of the interplay between the two gene expression machineries. Our findings provide a framework for future genetic, biochemical, and biophysical studies of coupled transcription and translation, and may guide the design of antimicrobials that uncouple these processes.

## Materials and methods

### Preparation of *E. coli* 30S•RNAP complex

70S ribosomes were prepared from *E. coli* (MRE600) as described (*Moazed and Noller, 1986*; *1989*). Ribosomal 30S and 30SΔS1 subunits were purified using sucrose gradient (10%–35%) fractionation in the ribosome dissociation buffer (20 mM Tris-HCl, pH 7.0; 300 mM NH$_4$Cl; 1.5 mM MgCl$_2$; 0.5 mM EDTA; 6 mM β-mercaptoethanol) and 30S-S1 dissociation buffer (20 mM Tris-HCl, pH 7.0; 1.5 M NH$_4$Cl; 10 mM MgCl$_2$; 0.5 mM EDTA; 6 mM β-mercaptoethanol), respectively. The fraction containing 30S or 30SΔS1 was concentrated and stored in ribosome-storage buffer (20 mM Tris-HCl, pH 7.0; 100 mM NH$_4$Cl; 10 mM MgCl$_2$; 0.5 mM EDTA; 6 mM β-mercaptoethanol) at –80°C. *E. coli* RNAP core with C-terminally 6 × His tagged β′-subunit was purified as described (*Nudler et al., 2003*), and stored at –80°C.

The 30S•RNAP complex was prepared for cryo-EM as follows: 30S subunit (1 μM final) was pre-activated at 42°C for 5 min in ribosome-storage buffer. Core RNAP (3 μM final) was added to the 30S solution and incubated for 30 min at 37°C. The same approach was used for the 30SΔS1•RNAP complex.

### RNAP binding assay

To analyze binding of RNAP core to 30S or 50S subunits or to 70S ribosome, 1 μM RNAP was mixed with purified 1 μM 30S subunit, 1 μM 50S subunit or 1 μM 70S ribosome (prepared by association of purified subunits) in ribosome-storage buffer (*Figure 1—figure supplement 1*) or in buffers with cell-like composition including glutamate and acetate anions (20 mM Tris-HCl, pH 7.0; 100 to 200 mM potassium glutamate; 10 mM ammonium acetate; 5 to 15 mM magnesium acetate; 2 mM spermidine) (*Jewett and Swartz, 2004*) (*Figure 1—figure supplement 2*). Each mixture was incubated for 30 min at 37°C, and 30 μl was layered over 10–30% linear sucrose gradients prepared in the corresponding buffer, in duplicate SW41 tubes (Beckman). Following centrifugation for 2 hr at 40,000 rpm, 4°C, three fractions were collected from each tube: a lighter fraction, peak fraction (30S or 50S or 70S), and heavier fraction (*Figure 1—figure supplements 1–3*). The presence of RNAP in each fraction was measured by dot-immunoblot assay (dot blot) using an HRP-conjugated anti-His antibody (GE Healthcare Life Sciences). 2 μl of each fraction was spotted onto nitrocellulose membrane in 0.5 μl steps. The membrane was dried at room temperature for 30 min, and blocked for 12 hr at 4°C in 20 ml PBS-T [phosphate-buffered saline, pH 7.0, containing 0.05% (v/v) Tween 20 (VWR, Inc.)] and 10% (w/v) dried non-fat milk (Labscientific, Inc.). The membrane was washed quickly in 20 ml of PBS-T, and then probed with HRP-linked anti-His antibody (1:4000 in 20 ml PBS-T) for 1 hr at room temperature. The membrane was washed 3 times with 20 ml PBS-T. 700 μl of freshly mixed Amersham ECL Prime Western Blotting Detection Reagent (GE Healthcare Life Sciences) was applied to the nitrocellulose membrane for 30 s. The membrane was dried and imaged using a Chemi Doc XRS + imaging system (BioRAD, Inc.) in chemi-resolution mode for 30 s.

To estimate the K$_d$ of the 30S•RNAP complex by the sucrose-gradient band sedimentation method (*Draper and von Hippel, 1979*), 50 nM RNAP was mixed with 2.5 μM 30S subunit in 20 mM Tris-HCl (pH 7.0), 150 mM potassium glutamate, 10 mM ammonium acetate, 5 mM magnesium acetate, 2 mM spermidine. 30 μl of the mixture was layered over 10–30% linear sucrose gradient prepared using the same buffer, in triplicate SW41 tubes (Beckman). Fractions were collected as described above (the 30S peak centered at 1.1 ± 0.05 cm), and 15 μl of each fraction was spotted onto nitrocellulose membrane and imaged as described above. The unbound and bound fractions of RNAP—that is, before the 30S subunit peak and at the 30S subunit peak, respectively—were quantified by subtracting the background of the buffer or the 30S subunit (Controls in *Figure 1—figure supplement 3*) from the signal (30S•RNAP data in *Figure 1—figure supplement 3*), using Image Lab software (version 5.2, BioRAD, Inc.). The Kd estimated from the triplicate experiments is $2.1 \times 10^{-8}$ M $\pm 6.3 \times 10^{-9}$ M.

## Analysis of 30S•RNAP interactions by crosslinking and mass-spectroscopy

### Crosslinking of 30S•RNAP complex with bis-sulfosuccinimidyl suberate (BS3)

Purified 30S subunit (3 μM) was combined with RNAP core (9 μM), and incubated for 10 min at room temperature. The complex was purified on 1 × 30 cm Superose 6 Increase column (GE LifeSciences) equilibrated in amine-free buffer: 20 mM HEPES-KOH, pH 7.5; 150 mM KCl; 10 mM MgCl$_2$; 0.05% β-mercaptoethanol. Peak fractions corresponding to 30S•RNAP complex were pooled, concentrated to ~10 μM 30S ribosomal subunit, and stored at 4°C.

Crosslinking was performed in 50 μl reactions containing 1 μM 30S•RNAP complex and 125 μM to 1 mM BS3 cross-linker (ThermoFisher, Waltham, MA USA). BS3 was dissolved in the gel-filtration buffer immediately before adding to the samples. Reactions were incubated for 30 min at room temperature, and quenched by mixing in 5 μl 1 M Tris-HCl, pH 7.5.

BS3-treated samples were mixed with LDS buffer (ThermoFisher), incubated at 75°C for 10 min, and fractionated on 4–12% NuPAGE gels using MES running buffer (ThermoFisher) at 200 V for 20 min. Gels were stained with Coomassie G-250 (SimpleBlue SafeStain, ThermoFisher), and bands that migrated slower than RNAP ββ' were excised from gel and analyzed by mass-spectroscopy to identify crosslinked peptides.

### LC-MS/MS and X-link mapping

Samples were reduced with 50 mM TCEP (all reagents were ThermoFisher Scientific LC-MS grade, unless indicated otherwise) for 10 min at 60°C, and alkylated with 50 mM iodoacetamide in the dark for 60 min at room temperature. In-gel digestion was performed at 37°C overnight with 0.5 μg sequencing-grade modified trypsin (Promega) and 100 mM ammonium bicarbonate. Resulting peptides were captured on C18 Spin Tips (Thermo Scientific) and eluted with 40 μl of 80% acetonitrile (ACN) in 0.1% trifluoroacetic acid. Eluted peptides were dehydrated in vacuum and suspended in 20 μl 0.1% formic acid for MS analysis.

Peptides were analyzed in the Orbitrap Fusion Lumos mass spectrometer (Thermo Scientific) coupled to an EASY-nLC (Thermo Scientific) liquid chromatography system. The peptides were eluted over a 120 min linear gradient from 96% buffer A (water) to 40% buffer B (ACN) followed by 98% buffer B over 20 min with a flow rate of 250 nl min$^{-1}$. Each full MS scan (R = 60,000) was followed by 20 data-dependent MS/MS (R = 15,000) with HCD and an isolation window of 2.0 *m/z*. Normalized collision energy was set to 35. Precursors of charge state 4 to 6 were collected for MS2 scans; mono-isotopic precursor selection was enabled, and a dynamic exclusion window was set to 30.0 s.

Raw LC-MS/MS data files were converted into *mgf* format and searched using pLink with default FDR < 5%, e-value set at <1, trypsin digest with up to three missed cleavages, constant modification at 1 = carbamidomethyl[C], variable at 1 = oxydation[M]. Cross-linker was set to BS3 ([K [K 138.068 138.068 156.079 156.079). Mass tolerances were left unaltered (default). *mgf* files were searched against database comprising all the *fasta* sequences of *E. coli* RNA polymerase core and 30S ribosome subunits.

## Cryo-EM and image processing

Holey-carbon grids (C-flat 1.2/1.3) were coated with a thin layer of carbon and glow discharged at 25 mA with a negative polarity setting for 45 s in an EMITECH K100X unit. Before application to the grids, the 30S•RNAP complex was diluted in ribosome-storage buffer to the following final concentrations: 50 nM 30S and 150 nM RNAP. 2.5 μl of the diluted sample was applied to the grids. After a 30 s incubation, grids were blotted for 5 s at blotting power 8, 10°C and ~95% humidity, and then plunged into liquid ethane using an FEI Vitrobot MK4. The grids were stored in liquid nitrogen. The same procedure was performed to prepare the 30SΔS1•RNAP sample grids.

A dataset of 184,530 30S•RNAP particles was collected as follows: 2527 movies were collected using SerialEM (*Mastronarde, 2005*) on an FEI Krios microscope operating at 300 kV, and equipped with a K2 Summit direct electron detector (Gatan Inc.) with −0.8 to −3.0 μm defocus. 50 frames per movie were collected at the total dose of 40 e-/Å$^2$ on the sample. The super-resolution pixel size was 0.82 Å on the sample.

A dataset of 272,975 30SΔS1•RNAP particles was collected as follows: 1641 movies were collected using SerialEM (*Mastronarde, 2005*) on an FEI Talos Arctica microscope operating at 200 kV,

and equipped with a K2 Summit direct electron detector (Gatan Inc.) with −0.8 to −3.0 µm defocus. 80 frames per movie were collected at the total dose of 40 e-/$Å^2$ on the sample. The super-resolution pixel size was 0.94 Å on the sample.

Particles for both datasets were extracted from aligned movie sums as follows: Movies were processed using IMOD (RRID:SCR_003297) (*Kremer et al., 1996*) to decompress frames and apply the gain reference. Movies were drift-corrected using unblur (*Grant and Grigorieff, 2015b*). Magnification anisotropy of the movie sums was corrected with mag_distortion_estimate and mag_distortion_correct (*Grant and Grigorieff, 2015a*). CTFFIND4 (*Rohou and Grigorieff, 2015*) was used to determine defocus values. 416 movies of 30S•RNAP complex (199 movies of 30SΔS1•RNAP complex) with high drift, low signal, heavy ice contamination, or very thin ice were excluded from further analysis after inspection of image sums and power spectra from CTFFIND4. Particles were automatically picked from full-sized images using cisTEM, software that implements a template-based algorithm (*Liu and Sigworth, 2014*). 560 × 560 pixel boxes with particles were extracted from super-resolution images, and the stack and FREALIGN parameter file were exported from cisTEM. To speed up processing, 2×- and 4×-binned image stacks were prepared using resample.exe, which is part of the FREALIGN distribution.

FREALIGN v9 (version 9.11) was used for all steps of refinement and reconstruction (*Grigorieff, 2016*) (*Figure 1—figure supplement 4*). The 4×-binned image stack was initially aligned to a 30S reference map, calculated from PDB 5U9F (*Demo et al., 2017*), and low-pass filtered to 20 Å. The resulting reconstruction with RNAP density was used to derive a 30S•RNAP reference (PDB 3LU0) (*Opalka et al., 2010*), and low-pass filtered to 20 Å for the following steps. Five rounds of mode 3 (global search) alignment included data in the resolution range from 30 Å to 300 Å. Next, the 4×-binned image stack was aligned against the common reference using mode 1 (local refinement), including data up to a high-resolution limit of 12 Å (20 Å in case of 30SΔS1•RNAP).

The refined parameters were then used for classification and refinement of the 2x-binned stack into 8 classes in 40 rounds, using resolutions from 12- to 300 Å, to separate the RNAP-containing classes. This step was done with three-dimensional (3D) masks that included RNAP with or without S1. In both cases, the resulting 30S•RNAP classes contained prominent S1 density. The 3D masks were created using IMOD and EMAN2 (*Tang et al., 2007*) by generating a density map, low-pass filtered to 20 Å, from our initial atomic models of RNAP with or without S1 (RNAP mask was used for 30SΔS1•RNAP data). The mask was applied to reference volumes in FREALIGN, with the volume outside of the mask weighted by a factor of 1.0 (*Grigorieff, 2016*). A five-pixel cosine edge was used on the mask and the masking filter function. This classification for the 30S•RNAP data set revealed 5 classes containing 30S and RNAP and 2 classes of free 30S subunit. For the classes bound with RNAP, particles with >50% occupancy were extracted from the 2×-binned stack. The new stack was sub-classified and refined into 4 classes in 60 rounds with the 3D mask, including RNAP and S1 (outside of the mask was down-weighted by a factor of 0.1) to resolve the RNAP, using data between 12- and 300 Å resolution. This classification resulted in 6.7- and 7.9 Å resolution maps (FSC = 0.143) for the non-rotated and rotated 30S•RNAP structures.

Classification of 30SΔS1•RNAP data into ten classes yielded two classes with defined features (~8.0 Å resolution) and eight classes at lower resolution (>12 Å). All maps contain RNAP bound similarly to that in the 30S•RNAP maps, and lacked the density ascribed to S1 in the 30S•RNAP maps. Particles with >50% occupancy in the higher-resolution classes were extracted from the 2×-binned stack. The new stack was sub-classified into 2 classes and refined in 40 rounds with the 3D mask (density outside the mask was down-weighted by a factor of 0.5), using data between 20- and 300 Å resolution. This resulted in 7.5 Å and 8.1 Å resolution maps that show similar RNAP positions, with a lower occupancy of RNAP in the 7.5 Å resolution map. The 8.1 Å map was used for structural comparisons in this work.

The maps used for structure refinements were sharpened using B-factors of −200 $Å^2$ to −250 $Å^2$ using bfactor.exe (included with the FREALIGN distribution) (*Grigorieff, 2016*). FSC curves were calculated by FREALIGN for even and odd particle half-sets.

## Model building and refinement

The 30S subunit from the 3.2 Å cryo-EM structure of *E. coli* 70S ribosome (PDB 5U9G) (*Demo et al., 2017*) and RNAP core subunits from high-resolution RNAP crystal structures (PDB 5UI8 and 5UAC) (*Campbell et al., 2017*; *Molodtsov et al., 2017*) were used as a starting model for structure

**Table 2.** Cryo-EM data collection and refinement statistics for 30S•RNAP structures

| | non-rotated | rotated | ΔS1 |
|---|---|---|---|
| PDB code | 6AWB | 6AWC | 6AWD |
| EMDB code | EMD-7014 | EMD-7015 | EMD-7016 |
| Data collection | | | |
| EM equipment | FEI Titan Krios | FEI Titan Krios | FEI Talos Arctica |
| Voltage (kV) | 300 | 300 | 200 |
| Detector | K-2 | K-2 | K-2 |
| Pixel size (Å) | 0.82 | 0.82 | 0.94 |
| Electron dose (e$^-$/Å$^2$) | 40 | 40 | 40 |
| Defocus range (μm) | −0.8 to −3.0 | −0.8 to −3.0 | −0.8 to −3.0 |
| Reconstruction | | | |
| Software | Frealign v9.11 | Frealign v9.11 | Frealign v9.11 |
| Number of particles used | 15,012 | 10,090 | 21123 |
| Final resolution (Å) | 6.7 | 7.9 | 8.1 |
| Map-sharpening $B$ factor (Å$^2$) | −198 | −205 | −200 |
| Model composition | | | |
| Non-hydrogen atoms | 75316 | 75169 | 76452 |
| Protein residues | 5743 | 5726 | 5573 |
| RNA bases | 1443 | 1443 | 1539 |
| Refinement | | | |
| Software | RSRef and Phenix | RSRef and Phenix | RSRef and Phenix |
| Correlation Coeff (%; Phenix) | 75.9 | 76.7 | 78.3 |
| R-factor (RSRef) | 0.250 | 0.251 | 0.257 |
| Validation (proteins) | | | |
| MolProbity score | 2.70 | 2.60 | 2.67 |
| Clash score, all atoms | 20.25 | 18.90 | 20.47 |
| Ramachandran-plot statistics (%) | | | |
| Favored (overall) | 80.0 | 81.9 | 79.6 |
| Allowed (overall) | 16.2 | 14.6 | 16.3 |
| Outlier (overall) | 3.8 | 3.5 | 4.1 |
| R.m.s. deviations | | | |
| Bond length (Å) | 0.005 | 0.004 | 0.004 |
| Bond angle (°) | 1.097 | 1.111 | 1.027 |
| Validation (RNA) | | | |
| Correct sugar puckers (%) | 99.9 | 99.9 | 99.9 |
| Good backbone conformation (%) | 88.2 | 88.6 | 88.2 |

DOI: https://doi.org/10.7554/eLife.28560.016

refinement. Ribosomal protein S2 and the N-terminal helix of S1 (aa 1–20) were built using the S1-S2 crystal structure (PDB 4TOI) (*Byrgazov et al., 2015*). The structures of domains 2 and 3 of S1 were built using PDB 2MFL (*Giraud et al., 2015*) and PDB 2KHI (*Salah et al., 2009*) by I-TASSER (RRID: SCR_014627) (*Yang et al., 2015*), and modeled as poly-alanine chains. Initial protein and ribosome domain fitting into cryo-EM maps was performed using Chimera (RRID:SCR_004097) (*Pettersen et al., 2004*), followed by manual modeling using Pymol (RRID:SCR_000305) (*DeLano, 2002*). The structural elements that were not defined by the cryo-EM maps were excluded from structural models.

Structural models were conservatively refined by real-space torsion-angle refinement using atomic electron scattering factors in RSRef (*Chapman, 1995*; *Korostelev et al., 2002*), essentially as described (*Svidritskiy et al., 2014*). Secondary-structure restraints, comprising hydrogen-bonding restraints for ribosomal proteins and base-pairing restraints for 16S rRNA, were employed as described (*Korostelev et al., 2008*). Refinement parameters, such as the relative weighting of stereochemical restraints and experimental energy term, were optimized to produce the stereochemically optimal models that closely agree with the corresponding maps. In the final stage, the structures were refined using phenix.real_space_refine (RRID:SCR_014224) (*Adams et al., 2011*), followed by a round of refinement in RSRef applying harmonic restraints to preserve protein backbone geometry and B-factor refinement in phenix.real_space_refine. The refined structural models have low real-space R-factors of 0.250, 0.251 and 0.257 for the non-rotated 30S•RNAP, rotated 30S•RNAP and 30SΔS1•RNAP structures, respectively. The resulting models have low deviation from ideal bond lengths and angles, low number of protein-backbone outliers and other robust structure-quality statistics, as shown in *Table 2*. Structure quality was validated using MolProbity (RRID:SCR_014226) (*Chen et al., 2010*).

The area of contact between RNAP and the 30S subunit was analyzed and calculated by the software PISA (CCP4 package) (*Krissinel and Henrick, 2007*; *Winn et al., 2011*). Structure superpositions and distance calculations were performed in Pymol. The cryo-EM maps for non-rotated (EMD-7014), rotated 30S•RNAP (EMD-7015) and 30SΔS1•RNAP (EMD-7016) structures were deposited in the EMDB (RRID:SCR_006506). PDB coordinates for non-rotated (PDB 6AWB), rotated (PDB 6AWC) 30S•RNAP and 30SΔS1•RNAP (PDB 6AWD) structures were deposited in the RCSB (RRID:SCR_012820). Figures were prepared in Pymol and Chimera (*DeLano, 2002*; *Pettersen et al., 2004*).

## Acknowledgements

We thank Zixuan Li for assistance with XL-MS; Zhiheng Yu and Chuan Hong for assistance with data collection at Janelia Research Campus; Chen Xu for help with preparing and screening cryo-EM grids at the cryo-EM facility at Brandeis University and for assistance with data collection at UMass Medical School; Darryl Conte Jr. and members of the Nudler and Korostelev laboratories for helpful comments on the manuscript. This study was supported by Howard Hughes Medical Institute (N G and E N) and NIH Grants R01 GM106105, GM107465 (to AAK), and GM107329 (EN).

## Additional information

### Competing interests

Nikolaus Grigorieff: Reviewing editor, *eLife*. The other authors declare that no competing interests exist.

### Funding

| Funder | Grant reference number | Author |
|---|---|---|
| National Institutes of Health | GM107329 | Evgeny Nudler |
| Howard Hughes Medical Institute | | Evgeny Nudler |
| National Institutes of Health | GM106105 | Andrei A Korostelev |
| National Institutes of Health | GM107465 | Andrei A Korostelev |

The funders had no role in study design, data collection and interpretation, or the decision to submit the work for publication.

### Author contributions

Gabriel Demo, Prepared ribosome-polymerase complexes, performed biochemistry, collected and processed cryo-EM data. Built and refined structural models. Wrote the manuscript; Aviram Rasouly, Nikita Vasilyev, Prepared ribosome-polymerase complexes, established 30S•RNAP binding and

performed XL-MS; Vladimir Svetlov, Analyzed XL-MS data; Anna B Loveland, Ruben Diaz-Avalos, Nikolaus Grigorieff, Assisted with cryo-EM data collection and data processing; Evgeny Nudler, Designed the project. Wrote the manuscript; Andrei A Korostelev, Designed the project. Built and refined structural models. Wrote the manuscript

### Author ORCIDs
Gabriel Demo http://orcid.org/0000-0002-5472-9249
Nikolaus Grigorieff http://orcid.org/0000-0002-1506-909X
Evgeny Nudler https://orcid.org/0000-0002-8811-3071
Andrei A Korostelev http://orcid.org/0000-0003-1588-717X

### Decision letter and Author response
Decision letter https://doi.org/10.7554/eLife.28560.030
Author response https://doi.org/10.7554/eLife.28560.031

# Additional files

### Supplementary files
• Transparent reporting form
DOI: https://doi.org/10.7554/eLife.28560.017

### Major datasets
The following datasets were generated:

| Author(s) | Year | Dataset title | Dataset URL | Database, license, and accessibility information |
|---|---|---|---|---|
| Demo G, Rasouly A, Vasilyev N, Svetlov V, Loveland AB, Diaz-Avalos R, Grigorieff N, Nudler E, Korostelev AA | 2017 | Structure of 30S ribosomal subunit and RNA polymerase complex in non-rotated state | https://www.rcsb.org/pdb/search/structid-Search.do?structureId=6AWB | Publicly available at the RCSB Protein Data Bank (accession no. 6AWB) |
| Demo G, Rasouly A, Vasilyev N, Svetlov V, Loveland AB, Diaz-Avalos R, Grigorieff N, Nudler E, Korostelev AA | 2017 | cryo-EM map of 30S ribosomal subunit and RNA polymerase complex in non-rotated state | https://www.ebi.ac.uk/pdbe/entry/emdb/EMD-7014 | Publicly available at the EMDataBank (accession no. EMD-7014) |
| Demo G, Rasouly A, Vasilyev N, Svetlov V, Loveland AB, Diaz-Avalos R, Grigorieff N, Nudler E, Korostelev AA | 2017 | Structure of 30S ribosomal subunit and RNA polymerase complex in rotated state | https://www.rcsb.org/pdb/search/structid-Search.do?structureId=6AWC | Publicly available at the RCSB Protein Data Bank (accession no. 6AWC) |
| Demo G, Rasouly A, Vasilyev N, Svetlov V, Loveland AB, Diaz-Avalos R, Grigorieff N, Nudler E, Korostelev AA | 2017 | cryo-EM map of 30S ribosomal subunit and RNA polymerase complex in rotated state | https://www.ebi.ac.uk/pdbe/entry/emdb/EMD-7015 | Publicly available at the EMDataBank (accession no. EMD-7015) |
| Demo G, Rasouly A, Vasilyev N, Svetlov V, Loveland AB, Diaz-Avalos R, Grigorieff N, Nudler E, Korostelev AA | 2017 | Structure of 30S (S1 depleted) ribosomal subunit and RNA polymerase complex | https://www.rcsb.org/pdb/search/structid-Search.do?structureId=6AWD | Publicly available at the RCSB Protein Data Bank (accession no. 6AWD) |
| Demo G, Rasouly A, Vasilyev N, Svetlov V, Loveland AB, Diaz-Avalos R, Grigorieff N, Nudler E, Korostelev AA | 2017 | cryo-EM map of 30S (S1 depleted) ribosomal subunit and RNA polymerase complex | https://www.ebi.ac.uk/pdbe/entry/emdb/EMD-7016 | Publicly available at the EMDataBank (accession no. EMD-7016) |

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
