## [Decision Letter]

[Editors’ note: this article was originally rejected after discussions between the reviewers, but the authors were invited to resubmit after an appeal against the decision.]

Thank you for submitting your work entitled "Structure of RNA polymerase bound to ribosomal 30S subunit" for consideration by *eLife*. Your article has been reviewed by three peer reviewers, one of whom is a member of our Board of Reviewing Editors and the evaluation has been overseen by a Senior Editor. The following individuals involved in review of your submission have agreed to reveal their identity: Alexander Mankin (Reviewer #3).

We have received comments from three expert reviewers all of whom found the data in the manuscript potentially interesting but, in light of the surprising configuration of RNAPol relative to the 30S subunit, and relative to the recently published work from the Cramer lab, to require further justification for its relevance. While the structure from the Cramer lab included RNAPol bound to the mRNA entry side of the small subunit of the ribosome, where previously implicated helicase proteins S3, S4, S5 are found, the structure here includes RNAPol bound to 30S ribosome subunits, on the exit side of the mRNA channel, where the authors argue that ribosomal protein S1 facilitates identification of SD motifs through interations with the aSD motif in the ribosome. Importantly, the Cramer structure represents a trapped native complex from a coupled in vitro transcription/translation reaction, whereas the structures in the Korostelev study derive from an in vitro binding reaction (corroborated by mass spec and sucrose gradient sedimentation analysis). Other minor criticisms include the observation of a dimer of RNAPol that may result from low salt conditions in the complex assembly reaction which might perturb any natural state of binding between 30S and RNAPol. In light of these major concerns, the reviewers found the manuscript to be unsuitable for publication in *eLife* at this stage.

*Reviewer #1:*

This manuscript by Korostelev and colleagues describes a cryoEM structure (at a resolution of about 7 Å) of bacterial RNA polymerase bound to a 30S bacterial ribosome small subunit. Using purified components, the authors first show using simple sucrose gradient fractionation that RNAP sediments preferentially to the 30S subunit, relative to the 50S and 70S complexes. And these results are confirmed by chemical cross-linking experiments showing interactions between B and b- subunits and small subunit proteins S1, S2, S18 and S21. With this knowledge, the authors determine structures of 30S subunits bound to RNAP by cryoEM which reveals an interaction consistent with the mass spec data where RNAP docs near the mRNA-binding site of the 30S subunit between the head and platform domains. Interestingly, the authors observe a second RNAP monomer more than 50 Å away from the primary binding site, but consistent with previous studies documenting the existence of RNAP dimers. The remainder of the Abstract details the structural observations of the complex which effectively place RNAP in a position to interact with the Shine-Dalgarno binding site of the 30S subunit where decisions about translation initiation must be made. Protein S1 density can be modeled in these structures in this same region, consistent with earlier studies, and this density is lost when ribosomes lacking S1 are purified. Classification reveals two different conformations of RNAP related to one another by a 7Å rotation, a movement that the authors argue opens the nucleic acid binding cleft of RNAP, but both structures are intermediate between the known open form of *E. coli* RNAP and the closed σ-bound holoenzyme form. The authors spend some time discussing how the RNA exit region of RNAP connects nicely to the 3' end of 16S rRNA (where the aSD element is found), and thus how this complex could rationalize the first step of translation initiation on an elongating RNAP complex. Overall, this is an intriguing new structure that likely reveals critical new insights into the long suspected coupling of RNAP elongation with translation initiation in bacteria.

Importantly, however, this structure follows on a recent publication from the Cramer group that has seemingly captured a very different complex – one between RNAP and the elongating 70S ribosome, thus presumably at a distinct stage downstream from the one captured here. Indeed, the structures are very different, with the interaction region in that structure found almost 80Å away from the binding site observed here, on the other side the small subunit where the proteins S3, S4, S5 implicated in mRNA unwinding are found. Given the very different stages which these two different structures are thought to capture, it is very possible that both are correct. But, critically, the Cramer group gives more confidence in the authenticity of the structure as it was formed in an actively transcribing and translating extract – these are stalled active complexes. By contrast, this study relies entirely on reconstituted biochemistry to capture relevant interactions.

Reviewer #2:

Transcription-translation coupling is acknowledged to be an important feature of the regulation of gene expression in bacteria. Here, the authors provide ~7 Å resolution cryo-EM structures of Eco core RNA polymerase (RNAP) bound to the small ribosomal 30S subunit. The manuscript has strengths and weaknesses:

Strengths:

The in vitro structural results are corroborated with in vitro binding (Figure 1—figure supplement 1) and crosslinking/mass-spectromety (Table 1) approaches. This is in contrast to a recent publication describing a cryo-EM structure of an RNAP ternary elongation complex (TEC) with a 70S ribosome (Kohler et al., 2017).

Weaknesses:

Presumably the substrate for transcription-translation coupling is the RNAP ternary-elongation complex (TEC), not the core RNAP. It's not completely clear why a TEC/30S complex was not investigated (rather than core RNAP/30S subunit).

The structural/biochemical results are not verified in vivo with, for example, mutagenesis studies.

The complex appears to contain one 30S ribosomal subunit but, for the most part, two molecules of core RNAP (one directly bound to the 30S and well resolved, the other dimerized with the 30S-bound RNAP and poorly resolved; Figure 1—figure supplement 2). This is likely due to the relatively low salt concentration of the buffer (*E. coli* core RNAP is well known to dimerize at Cl^-^ ion concentrations below ~250 – 300 mM; for instance, see Shaner et al., 1982). The conformational states of the directly bound RNAP are thus strongly influenced by the dimer formation and not likely to be relevant to 30S binding, etc.

The manuscript lacks a discussion of transcription elongation factors, such as NusG, known to play an important role in transcription-translation coupling

Overall, I feel that the major weaknesses outweigh the strength and need to be addressed before publication of this manuscript.

Reviewer #3:

Demo et al. examined the structure of the *E. coli* 30S ribosomal subunit complexed to the core RNA polymerase. After forming in vitro the complex the authors solved its structure by cryo-EM at a ~ 7 Å resolution. The structure places the RNAP RNA exit site close to the 16S rRNA anti-Shine Dalgarno region. An idiosyncratic density was identified as representing two OB domains of the ribosomal protein S1.

Critique:

The work is innovative and generally interesting. However, there are several important questions, which are not sufficiently addressed. Furthermore, the recent publication of the cryo-EM structure of the 70S ribosome complexed with RNAP ("expressome') (Science, 356, 194-197) requires from the authors of this paper more detailed comparison of the 30S-RNAP structure with the published model and a more rigorous addressing of the differences between the two structures.

Major points:

1) From the fairly congested images shown it the figures, it remains unclear (at least to me) whether the placement of RNAP on the 30S subunit would clash if the 50S subunit would bind. If it does not clash, how do authors explain the lack of interaction of RNAP with the 70S ribosome? If it does clash, what is the proposed scenario of the 50S binding?

2) In the published structure of the 'expressome', the RNA exit tunnel of RNAP aligns with the mRNA channel entrance site on the ribosomal 30S subunit. This seems to make biological sense. In contrast, in the structure of Demo et al., the RNAP RNA exit 'is placed next to the Shine-Dalgarno binding site', meaning near the mRNA channel exit. seems to align with the 30S mRNA exit (the 16S 3' end). I am not sure I fully understand how do authors envision the mRNA trajectory from RNAP to the 30S mRNA channel entrance.

3) The work needs stronger evidence that the visualized complex is biologically relevant and not just an aberrance resulting from the artificial in vitro conditions of the complex formation.

[Editors’ note: what now follows is the decision letter after the authors submitted for further consideration.]

Thank you for choosing to send your work entitled "Structure of RNA polymerase bound to ribosomal 30S subunit" for consideration at *eLife*. Your letter of appeal has been considered by a Senior Editor and a Reviewing editor, and we are prepared to consider a revised submission with no guarantees of acceptance.

In the revised submission, please objectively address the published expressome structure and its relationship to the current structure (which was absent from the previous manuscript) and also address the feature of your structural model wherein the RNAP RNA exit channel is located near the 30S mRNA exit tunnel (i.e. why would the two exit tunnels be aligned?). Better figures will be needed to address this latter point – perhaps accompanied by clear cartoons.

---

## [Author Response]

[Editors’ note: the authors’ letter of appeal follows.]

We are encouraged by the reviewers’ enthusiasm about our work, stating that: “this is an intriguing new structure that likely reveals critical new insights” (reviewer 1), “structural results are corroborated with in vitro binding and crosslinking/mass-spectrometry…in contrast to a recent publication describing a cryo-EM structure of an RNAP ternary elongation complex (TEC) with a 70S ribosome (Kohler et al, 2017)” (reviewer 2), and that “this work is innovative and generally interesting” (reviewer 3).

We believe that our work meets *eLife*’s publication standards as it presents a new structural view of translation-transcription coupling that explains many previous results (see below), and it advances the field by enabling future mechanistic studies.

The reviewers expressed concerns that must be addressed prior to publication. They request that (1) we compare our structure to the recently published 70S*RNAP complex (“expressome”: (Kohler et al., 2017), and (2) we discuss the functional relevance of our structure in a cellular context, including the possibility that the complex we observe is an “in vitro” artifact. We would like to address the reviewers’ concerns in a revised manuscript.

We initially avoided a detailed comparison with the 70S*RNAP complex because, as noted by reviewer 1, we believe that our structure captures a different step in the transcription-translation pathway. Nevertheless, we have compared the structures as best we can, and we propose to describe the relevant features in the revised manuscript. Comparison of our structures with the 70S*RNAP complex indicates that our work provides a structurally and functionally realistic view.

As to the relevance of our structure, we have new data showing that the 30S*RNAP complex is stable under a wide range of conditions, including those that mimic conditions inside living cells and in vitro transcription translation systems. By contrast, the 70S*RNAP “expressome” structure (Kohler et al., 2017)—in which mRNA tethers stalled RNAP and the 70S ribosome—was obtained using glutaraldehyde cross-linking in a buffer without major cellular cations and anions (see below).

We’ve also performed a structure-based quantitation that supports the high stability of our complex. Systematic analyses show that most functional cellular complexes have contact areas from ~500 Å2 (weaker binding) to ~2,000 Å2 (strong binding) and binding affinities (K_d_) in the nanomolar-to-low-μM range. For example, the TGF-β*TGF-β receptor (K_d_ = 3 × 10^-7^) and Integrin α*C3D complexes (K_d_ = 5 × 10^-7^) have contact areas of ~1000 Å2. In our structures, the areas of contact between RNAP and 30S are ~1700 and ~1800Å2.

We have measured a K_d_ for our structure in cell-like conditions. The K_d_ of less than 1 × 10-7 M indicates high stability of the complex at cellular concentrations of free RNAP (~1 × 10-6 M – (Klumpp and Hwa, 2008) and (Bakshi et al., 2013) and ribosomes/subunits (50-100 × 10-6 M), consistent with the high contact area in our structures.

Evidence from the literature also points to the in vivo relevance of the 30S*RNAP complex (Sanamrad et al., 2014; Sukhodolets and Garges, 2003; Sukhodolets et al., 2006) in various bacteria (Buttner et al., 1997; Chakrabarti and Gorini, 1977; Karunakaran and Davies, 2000), and suggests that it could have at least two functions:

1) S1 interacts with RNAP and enhances transcription (Sukhodolets and Garges, 2003) and translation initiation (Sanamrad et al., 2014; Sukhodolets et al., 2006). The latter published works suggest that S1 directly links transcription with translation by loading mRNA onto the translation machinery.

2) S2 interacts with RNAP in *E. coli* (Sukhodolets and Garges, 2003) and in *Bacillus* (Buttner et al., 1997). In *Bacillus*, the interaction was dependent on heat shock, which causes dissociation of 70S ribosomes and accumulation of ribosomal subunits (Zhang et al., 2015). The 30S*RNAP complex could function to “colocalize” the transcription and translation machineries to efficiently resume gene expression as stress conditions subside.

These interactions are present in our complex, but not in the 70S*RNAP complex.

Typical in vivo experiments (e.g., mutagenesis experiments) to test the relevance of the interactions in our complex are not straightforward because the contact sites are essential for transcription and translation. The absence of in vivo data does not invalidate our conclusions. Rather, we conclude that the 30S*RNAP complex is perfectly rational, as it helps explain existing in vivo data, which show that transcription and translation are coupled and that ribosomal subunits and RNAP co-localize.

By presenting new data, citing published literature and expanding our discussion, we can address each point raised by the reviewers. Because all three reviewers agree that our structural findings are generally interesting and are corroborated by our own and published work, we would be grateful for an opportunity to submit a revised manuscript.

[Editors’ note: the author responses to the re-review follow.]

Thank you for choosing to send your work entitled "Structure of RNA polymerase bound to ribosomal 30S subunit" for consideration at eLife. Your letter of appeal has been considered by a Senior Editor and a Reviewing editor, and we are prepared to consider a revised submission with no guarantees of acceptance.In the revised submission, please objectively address the published expressome structure and its relationship to the current structure (which was absent from the previous manuscript) and also address the feature of your structural model wherein the RNAP RNA exit channel is located near the 30S mRNA exit tunnel (i.e. why would the two exit tunnels be aligned?). Better figures will be needed to address this latter point – perhaps accompanied by clear cartoons.

Thank you for the opportunity to submit a revised manuscript describing our cryo-EM structures of 30S*RNAP complexes. We’ve substantially improved the manuscript by showing that the 30S subunit and RNAP bind with high affinity under a wide range of ionic conditions, including those that approximate cellular conditions optimal for transcription-translation coupling (described in the Results section and shown in Figure 1—figure supplement 2 and Figure 1—figure supplement 3). We show that the large area of contact in our 30S*RNAP structures indicates functional interaction. We’ve added a discussion of the mechanistic link between our structure and the tethered 70S*RNAP “expressome” structure recently published by Kohler et al., (2017), providing additional figures and a cartoon schematic (Figure 4 and Figure 5), as you suggested. We illustrate that the structures are consistent with distinct steps of the transcription-translation coupling: co-transciptional 30S maturation, translation initiation (30S*RNAP) and elongation (70S*RNAP). We also point to additional published work that supports the relevance of our complex in the cellular environment. Moreover, we’ve updated the text, figures, and references to address each suggestion or concern raised by the reviewers.

As noted in our revised Discussion, the Blaha lab has just published an independent biochemical study (https://doi.org/10.1093/nar/gkx719) showing that RNAP binds with to 30S and 70S. Importantly, they identify the same binding interface that we report in our 30S*RNAP structure, despite using different cross-linking approaches and buffer conditions. The Blaha lab’s biochemical work complements our work and strengthens our conclusions, which emphasizes the importance of our description of the first 30S*RNAP structures.